# Thumb on the Scale: Optimal Loss Weighting in Last Layer Retraining

**Nathan Stromberg**
Arizona State University
`nstrombe@asu.edu`

**Christos Thrampoulidis**
University of British Columbia
`cthrampo@ece.ubc.ca`

**Lalitha Sankar**
Arizona State University
`lsankar@asu.edu`

## Abstract

While machine learning models become more capable in discriminative tasks at scale, their ability to overcome biases introduced by training data has come under increasing scrutiny. Previous results suggest that there are two extremes of parameterization with very different behaviors: the population (underparameterized) setting where loss weighting is optimal and the separable overparameterized setting where loss weighting is ineffective at ensuring equal performance across classes. This work explores the regime of last layer retraining (LLR) in which the unseen limited (retraining) data is frequently inseparable and the model proportionately sized, falling between the two aforementioned extremes. We show, in theory and practice, that loss weighting is still effective in this regime, but that these weights *must* take into account the relative overparameterization of the model.

## 1 Introduction

While discriminative machine learning has produced exceptional predictive power in many settings, even the most expressive models have struggled to balance the accuracy of common (majority) classes and rare (minority) classes. This comes as the most critical tasks, such as disease prediction, fraud detection, and rare event monitoring are fundamentally extremely imbalanced. It has, therefore, become critical to correct the existing majority bias even in the age of highly expressive, overparameterized models.

Substantial research has gone into studying methods for correcting models for their majority bias, and the most successful lines of research have been driven by the idea of last layer retraining (LLR) [1, 2] By updating only the final linear layer of a large model, the number of parameters to train is greatly reduced while still providing the flexibility to balance the accuracy for minority classes. In this setting, a variety of cost-sensitive methods have been investigated, including weighted empirical risk minimization (wERM) sometimes called importance weighting [3], downsampling [4, 3], and other corrected losses [5–10]. Each of these methods has shown compelling empirical success in the LLR setting, improving the accuracy of minority classes or groups to match that of majority groups. Despite this, theoretical and empirical evidence has shown that, for overparameterized models, wERM has no effect on the learned model whatsoever [11, 12]. How do we reconcile this seeming contradiction when we are correcting large models?

Last layer retraining is often characterized by two quantities, the latent data dimension $d$ (input to the last layer) and the number of retraining samples $n$. The overparameterization ratio $\delta \triangleq d/n \in (0, \infty)$ has been used to breakdown the performance of the above-mentioned methods into two critical regimes: the populations setting ($\delta \to 0$) and the overparameterized setting ($\delta > 1$). For example, in the population setting, the methods described above for LLR have demonstrated both empirical and theoretical success in the population regime [3]. On the other hand, only downsampling and CS-SVM have been successful in improving minority class accuracy when the model is highly overparameterized, i.e., $\delta > 1$, and can separate the data [4, 9, 10]. In this work, we

39th Conference on Neural Information Processing Systems (NeurIPS 2025).

consider the ratio $\delta$ in the understudied but highly relevant regime of $\delta \in (0, 1)$. This underparameterized regime is critical because, in practice, the number of samples is often on the same order as the number of parameters in the last layer retraining scenario.

In this underparameterized regime, we study the problem of obtaining loss weights which minimize worst-class error (WCE) for binary classification. Our key contributions in this setting are as follows:

- We simplify the wERM optimization with any loss on a sample from a class-conditional Gaussian distribution to a general system of four scalar equations for the setting where both $d, n \to \infty$

- For the setting above and the choice of square loss, we derive optimal asymptotic weighting as a function of the overparameterization ratio $\delta$.

- We compare this optimal weighting scheme to downsampling and show that optimal wERM outperforms downsampling, especially when data is limited.

- Finally, we show that the trends described for simple Gaussian data in theory appear in real-world image classification problems and that our optimal weighting scheme can outperform the classical ratio of priors by leveraging the notion of an effective (latent) dimension.

## 1.1 Related Work

**Importance Weighting in Separable Settings**  While wERM has been successful in practice, its applicability in high-dimensional problems has been called into question. Soudry et al. [13], Ji and Telgarsky [14], Gunasekar et al. [15] explore implicit bias of gradient descent-type methods on linear models and characterize the limiting model for a variety of loss families, including square and logistic losses. Xu et al. [16] explicitly consider loss weighting in the implicit bias and find that importance weighting on common losses does not alter the learned model, but can improve the convergence rate in imbalanced settings. Byrd and Lipton [11], Sagawa et al. [17] explore the problem of importance weighting with a focus on deep models and find that larger models are less affected by loss weighting than smaller models.

**Overparameterized Learning**  Kini et al. [9], Behnia et al. [18] study CS-SVM in the overparameterized regime and show that in this setting the ineffectiveness of loss weighting can be overcome by considering an altered loss, namely the vector scaling loss. Their theoretical results help to explain important behaviors for $\delta > 1$. Lyu et al. [8] study a similar loss for $\delta > 0$ and find that even when $\delta < 1$, the margin-scaled loss can be helpful for mitigating model biases. Lai and Muthukumar [10] study margin-adjusted minimum norm interpolation in the overparameterized regime and show that this can help to mitigate poor out-of-distribution generalization. Our work instead considers loss weighting (as opposed to margin weighting) in the underparameterized regime.

**Downsampling**  Chaudhuri et al. [4] study downsampling in the context of separable SVM and work with fixed $d$ while scaling up the class means as $n \to \infty$. Their work utilizes extreme value theory and finds that downsampling can improve WCE for several data distributions. Several works empirically utilize downsampling in the LLR setting to improve subgroup fairness including Kirichenko et al. [1], LaBonte et al. [19], Stromberg et al. [20]. Our work focuses weighted learning on inseparable dala and the proportional regime of $\delta \in (0, 1)$.

**Population Risk Minimization**  In the population setting, Welfert et al. [3] study wERM, downsampling, and MixUp. They show that the first two are equivalent problems, and that all three result in the same solution given $n \to \infty$. We instead focus on both $d, n \to \infty$ at a fixed ratio which can better explain practical settings like LLR where the number of data points and the number of trainable parameters are on the same order.

## 2 Problem Setup

We denote deterministic vectors as $\underline{x}$ and their random counterparts as $\underline{X}$. Matrices are denoted $\boldsymbol{X}$ and their randomness is clear by context. We begin by defining the generative model and key functions of interest.

## 2.1 Data and Metrics

We assume the following generative model for the latent data $\underline{X}$ and label $Y$:

$$Y \sim \begin{cases} -1 & \text{w.p. } \pi_- \\ 1 & \text{w.p. } \pi_+ \end{cases}$$

$$\underline{X}|(Y = y) \sim \mathcal{N}(y\underline{\mu}, \mathbf{I})$$

with $\underline{\mu} \in \mathbb{R}^d$ and $\mathbf{I} \in \mathbb{R}^{d \times d}$ the identity matrix. Without loss of generality, we assume that $\pi_+ < \pi_-$ so $+1$ is the minority class. Assume we get $n$ samples as $(\boldsymbol{X}, \underline{y}) \in \mathbb{R}^{n \times d} \times \mathbb{R}^n$ following the above distribution.

The key metric studied in this work is worst-class error (WCE) which provides a notion of fairness for a given classifier. We can calculate the expected risk (error) on each class for a given linear model $(\underline{\theta}, b)$ as follows:

$$\mathcal{R}_+ \triangleq Q\left(\frac{\gamma s}{\alpha} + \frac{b}{\alpha}\right), \qquad \mathcal{R}_- \triangleq Q\left(\frac{\gamma s}{\alpha} - \frac{b}{\alpha}\right), \tag{1}$$

where $s \triangleq \|\underline{\mu}\| \in \mathbb{R}$ is the signal strength, $\gamma \triangleq \frac{\mu^\top \theta}{\|\mu\|} \in \mathbb{R}$ is the energy of $\underline{\theta}$ along $\mu$, $\alpha \triangleq \|\underline{\theta}\| \in \mathbb{R}$ is the total energy of $\underline{\theta}$, and $Q(\cdot)$ is the standard $Q$-function. More details can be found in Appendix A.1. These three scalar quantities and the bias will be the focus of our analysis.

We can then calculate WCE as follows:

$$\text{WCE}(\underline{\theta}, b) \triangleq \max\{\mathcal{R}_+, \mathcal{R}_-\} \tag{2}$$

In the sequel, we will make use the following function class: the Moreau envelope of a function $f$ is parameterized by $\lambda$ and defined as

$$\mathcal{M}_f(x; \lambda) \triangleq \inf_v f(v) + \frac{1}{2\lambda}\|v - x\|_2^2 \tag{3}$$

and can be thought of as a locally smoothed version of $f$. It is always continuously differentiable with respect to $x$ and $\lambda$. The $v$ that achieves the infimum above is given by the prox operator. We will also make use of the partial derivatives of $\mathcal{M}$ which we denote as $\mathcal{M}'_{\ell,1}(x; \lambda)$ and $\mathcal{M}'_{\ell,2}(x; \lambda)$ for the partials with respect to the first and second arguments respectively. The second derivatives are denoted similarly. See Appendix A.3 for a full definition.

## 3 Asymptotic Analysis of Cost-Sensitive Last Layer Methods

We will now discuss our key theoretical results for both weighted and downsampled ERM in the asymptotic setting where both $d, n \to \infty$ such that $\delta = d/n \in (0, 1)$. Furthermore, we specialize in the analysis for square loss for which we can derive closed form solutions.

### 3.1 Weighted ERM

We consider the weighted ERM problem

$$\min_{\underline{\theta}, b} \frac{1}{n} \sum_{i=1}^n \omega_i \ell\left(y_i(\underline{x}_i^\top \underline{\theta} + b)\right) \tag{4}$$

where $\ell$ is the margin-based loss function of interest and is only required to be convex in its argument. We consider the special case of *class weighting* where, recalling that class $+1$ is the minority class, we have

$$\omega_i = \begin{cases} 1, & y_i = -1 \\ \rho, & y_i = +1. \end{cases} \tag{5}$$

Although weighting of groups or other subsets of the data fits into our framework, we focus on class weighting for clarity. We denote the solution to this problem by $(\hat{\underline{\theta}}(\rho), \hat{b}(\rho))$. Note that we can reparameterize this solution as $(\hat{\alpha}(\rho), \hat{\gamma}(\rho), \hat{b}(\rho))$ with $\alpha$ and $\gamma$ defined as in section 2.1.

We begin by showing how this $(d+1)$-dimensional optimization can be reduced to the solution of a system of scalar equations.

**Theorem 1** (Weighted ERM Solution). *For $\delta \in (0,1)$ and a proper, convex loss $\ell$, (4) can be effectively reduced to a four dimensional system of equations in $\alpha, \gamma, \lambda, b$ given by*

$$\delta(\alpha^2 - \gamma^2) + 2\lambda^2 \pi_+ \mathbb{E}[\mathcal{M}'_{\ell,2}(-\alpha G + s\gamma + b; \lambda)]$$
$$+ 2\lambda^2/\rho^2 \pi_- \mathbb{E}[\mathcal{M}'_{\ell,2}(-\alpha G + s\gamma - b; \lambda/\rho)] = 0 \tag{6a}$$

$$\frac{\delta \gamma \rho}{\lambda} + \pi_+ \rho s \mathbb{E}[\mathcal{M}'_{\ell,1}(-\alpha G + s\gamma + b; \lambda)]$$
$$+ \pi_- s \mathbb{E}[\mathcal{M}'_{\ell,1}(-\alpha G + s\gamma - b; \lambda/\rho)] = 0 \tag{6b}$$

$$-\frac{\delta \alpha \rho}{\lambda} + \pi_+ \rho \alpha \mathbb{E}[\mathcal{M}''_{\ell,1}(-\alpha G + s\gamma + b; \lambda)]$$
$$+ \pi_- \alpha \mathbb{E}[\mathcal{M}''_{\ell,1}(-\alpha G + s\gamma - b; \lambda/\rho)] = 0 \tag{6c}$$

$$\pi_+ \rho \mathbb{E}[\mathcal{M}'_{\ell,1}(-\alpha G + s\gamma + b; \lambda)]$$
$$-\pi_- \mathbb{E}[\mathcal{M}'_{\ell,1}(-\alpha G + s\gamma - b; \lambda/\rho)] = 0 \tag{6d}$$

*with $G \sim \mathcal{N}(0,1)$. Concretely, when the solution $(\alpha^*, \gamma^*, \lambda^*, b^*)$ to this system of equations is unique, then it satisfies $(\hat{\alpha}, \hat{\gamma}, \hat{b}) \to (\alpha^*, \gamma^*, b^*)$ in probability as $n, d \to \infty$ with $d/n = \delta$ fixed.*

The proof is presented in Appendix A.4 and relies on the convex Gaussian minimax theorem (CGMT) [21, 22] which is discussed in Appendix A.2 and further in [23].

The expected Moreau envelope of $\ell$ can be estimated from samples, allowing the system in (6) to be efficiently solved using numerical methods like fixed-point iteration. For square loss, however, the Moreau envelope admits a closed form

$$\mathcal{M}_{\ell_{\text{square}}}(x; \lambda) = \frac{1}{2} \frac{(x-1)^2}{1+\lambda}, \tag{7}$$

allowing us to reduce the above system and remove the randomness.

**Corollary 1** (Square Loss). *Letting $\ell(z) = \ell_{square}(z) \triangleq \frac{1}{2}(z-1)^2$, we have the following set of equations as a simplification of (6):*

$$\pi_+ \frac{\lambda^2}{(1+\lambda)^2}((s\gamma + b - 1)^2 + \alpha^2) + \pi_- \frac{\lambda^2}{(\rho+\lambda)^2}((s\gamma - b - 1)^2 + \alpha^2) = \delta(\alpha^2 - \gamma^2) \tag{8a}$$

$$\pi_+ s(s\gamma + b - 1)\frac{\lambda}{1+\lambda} + \pi_- s(s\gamma - b - 1)\frac{\lambda}{\rho+\lambda} = -\delta \gamma \tag{8b}$$

$$\pi_+ \frac{\lambda}{1+\lambda} + \pi_- \frac{\lambda}{\rho+\lambda} = \delta \tag{8c}$$

$$\pi_+ \frac{s\gamma + b - 1}{1+\lambda} - \pi_- \frac{s\gamma - b - 1}{\rho+\lambda} = 0 \tag{8d}$$

*and additionally this system has a unique solution.*

*Proof Sketch.* We use the closed form for the Moreau envelope of the square loss and evaluate all of the expectations utilizing the known data distribution. The uniqueness of the solution follows from the joint convexity of the combined expected Moreau envelopes. A full proof is provided in Appendix A.5. □

While this system of equations is significantly simpler than that of (6), it still does not admit a closed-form solution for general weighting. However, certain special cases including unweighted ERM allow for a closed-form solution. Substituting $\rho = 1$ in (8), we obtain the following corollary:

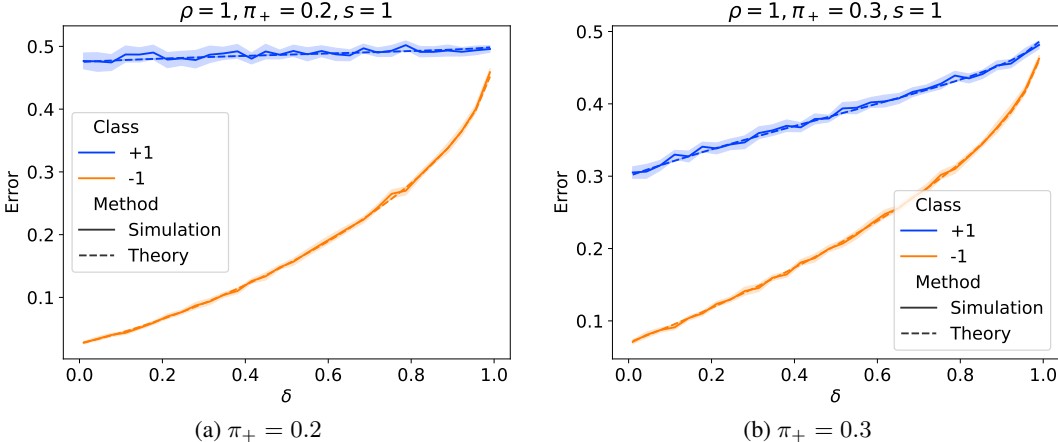

(a) $\pi_+ = 0.2$           (b) $\pi_+ = 0.3$

Figure 1: Plot of per-class test error as a function of $\delta$ on a class-conditional Gaussian dataset. We see that simulation matches the theoretical closed form across all $\delta$ values and different imbalances $\pi_+$. Both classes perform more poorly for growing overparameterization. The majority always outperforms the minority, however.

**Corollary 2** (Unweighted Solution). *Letting $\rho = 1$, we get the following solution to* (8):

$$\gamma^* = \frac{s(1 - (\pi_- - \pi_+)^2)}{1 + s^2(1 - (\pi_- - \pi_+)^2)} \tag{9a}$$

$$\lambda^* = \frac{\delta}{1 - \delta} \tag{9b}$$

$$b^* = (\pi_- - \pi_+)(\gamma^* s - 1) \tag{9c}$$

$$\alpha^* = \sqrt{\lambda^*(\pi_+(\gamma^* s + b^* - 1)^2 + \pi_-(\gamma^* s - b^* - 1)^2) + \frac{\gamma^{*2}}{1 - \delta}} \tag{9d}$$

Similar results have appeared in the literature [24], and we include it here for completeness.

The solution in Corollary 2 captures the effect of overparameterization through $\delta$. We visualize this effect in Figure 1 where we also see that the small $\delta$ regime strongly favors the majority whereas as $\delta \to 1$, the difference between classes dissipates (at the cost of increasing WCE). Its behavior as a function of $\pi_+$ is shown in Figure 4.

An interesting case is when we want to equalize the error of each class. From (1), we obtain this by setting $b = 0$; this, in turn, corresponds to choosing a specific $\rho$ which yields $b^* = 0$. Starting from Corollary 1, we obtain the following result for the equal error settings.

**Theorem 2** (Equal Error Solution). *Assuming $\delta < 2\pi_+$, let $\rho = \tilde{\rho}$ defined as*

$$\tilde{\rho} \triangleq \frac{\pi_-}{\pi_+} + \left(\frac{\pi_-}{\pi_+} - 1\right)\frac{\delta}{2\pi_+ - \delta} \tag{10}$$

*Then, the solution to* (8) *is given by*

$$b^* = 0 \tag{11a}$$

$$\gamma^* = \frac{s}{1 + s^2} \tag{11b}$$

$$\alpha^* = \sqrt{\frac{\Delta}{1 - \Delta}(\gamma^* s - 1)^2 + \frac{\gamma^{*2}}{1 - \Delta}}, \tag{11c}$$

*where we define $\Delta := \frac{\delta}{4\pi_+} + \frac{\delta}{4\pi_-}$ for convenience. We can write the worst-class error of wERM using $\tilde{\rho}$ and square loss as*

$$Q\left(\frac{s^2\sqrt{1 - \Delta}}{\sqrt{\Delta + s^2}}\right). \tag{12}$$

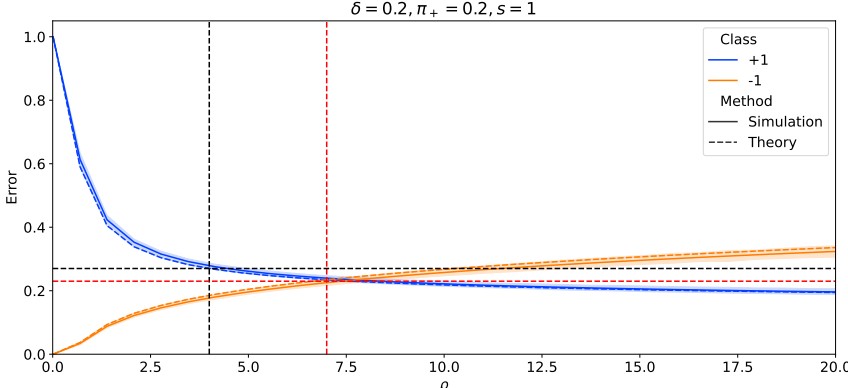

Figure 2: Per-class test error plotted against the weight ratio $\rho$ on a synthetic dataset. We see that not only does our theoretical formulation match the simulation results, $\tilde{\rho}$ (red) strongly outperforms the conventional ratio of priors (black). Note that the worst-class error for each weighting scheme is marked with a horizontal line.

*Proof.* Substituting $\tilde{\rho}$ into (8) we can solve for $\lambda, \gamma, b, \alpha$. To get the WCE we substitute into (1). $\quad\square$

We see that the form of $\tilde{\rho}$ is the classical ratio of priors plus an offset that depends only on the overparameterization ratio $\delta$ and the minority prior $\pi_+$. Note that this offset is 0 for balanced classes regardless of parameterization.

We see in Figure 2 that $\tilde{\rho}$ aligns with the crossover of the per-class errors in theory and in simulation of a class-conditional Gaussian with $s = 2, \delta = 0.2$, and $\pi_+ = 0.2$. Additionally, we see the monotonicity required by Theorem 3. This results in $\tilde{\rho}$ (red) strongly outperforming the population-optimal weighting marked in black. This suggests that using a stronger weight should be helpful in practical settings where $d$ and $n$ are on similar scales. We explore this further in section 4.

It is worth noting that this $\tilde{\rho}$ for WCE does not align with the conventional wisdom of the ratio of the priors except as $\delta \to 0$, which is the case that $n \to \infty$ for a fixed $d$. This aligns with previous population results such as [3]. We compare the ratio of the priors to $\tilde{\rho}$ in Figure 2 and indeed see that the ratio of the priors is suboptimal for this $\delta > 0$.

We additionally see in Theorem 2 that the solution $\tilde{\rho}$ is only valid in the setting that $\delta < 2\pi_+$. This is an interesting restriction, as it suggests that overparameterization affects the ability of weighting to correct for class imbalances. Indeed we see in Figure 3b that when $\delta > 2\pi_+$, the two per-class risks never intersect as a function of $\rho$, and thus the WCE is dominated only by the minority class. In this setting, the optimal choice of weighting is $\rho \to \infty$.

With the assumption that the per-class risks are monotonic in $\rho$ (which holds in our simulations), this choice of $\rho$ is optimal in terms of WCE. While the assumption above may seem strong, it is intuitive that increasing the weight on a class should decrease the risk for that class and increase the risk for the opposite class.

**Theorem 3** (Optimality of $\tilde{\rho}$). *Assume that $\mathcal{R}_+$ and $\mathcal{R}_-$ are monotonically decreasing and increasing respectively in the weight parameter $\rho$. Then $\tilde{\rho}$ minimizes WCE over all choices of $\rho$ for a given $\mu$ and $\delta < 2\pi_+$.*

*Proof.* For convenience, we overload WCE to be a function of weight $\rho$ directly. If both per-class risks are monotonic in $\rho$, WCE is a quasiconvex function in $\rho$ as it is the maximum of two monotonic functions. The choice of $\tilde{\rho}$ is a local minima of WCE as for any $\epsilon > 0$, $\text{WCE}(\tilde{\rho} + \epsilon) \geq \text{WCE}(\tilde{\rho})$ as $\mathcal{R}_-$ is increasing and $\text{WCE}(\tilde{\rho} - \epsilon) \geq \text{WCE}(\tilde{\rho})$ because $\mathcal{R}_+$ is decreasing. Thus $\tilde{\rho}$ is a global minima of WCE by quasiconvexity. $\quad\square$

## 3.2 Downsampled ERM

While upweighting is one common form of cost-sensitive correction during LLR, another is downsampling. We specifically consider the variant of downsampling where the majority class is down-

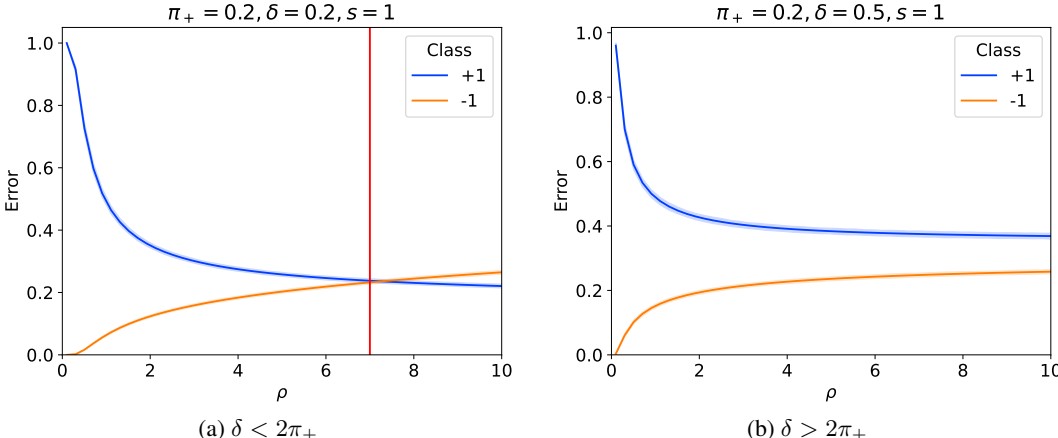

(a) $\delta < 2\pi_+$

(b) $\delta > 2\pi_+$

Figure 3: Effect of $\rho$ on the per-class test error on a synthetic dataset. We see the restriction for $\delta < 2\pi_+$ for $\tilde{\rho}$ (red) to be defined has an operational meaning regarding the per-class errors. Indeed if $\delta > 2\pi_+$, the per-class errors never meet and thus cannot be balanced.

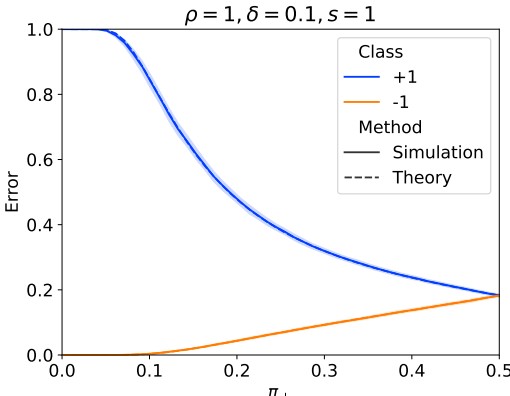

Figure 4: Per-class errors for unweighted ERM. We first note that simulation using SGD matches the theoretical risk across all $\pi_+$ values. The effect of imbalance is quite extreme and the minority error quickly skyrockets.

Figure 5: A comparison of wERM using the optimal weight $\tilde{\rho}$ and downsampling. We see that for a fixed $\delta$, the error for wERM is much lower than that of downsampling but that both achieve balanced per-class errors.

sampled to the size of the minority class. We can adapt our existing results to capture the downsampling problem as a change of $n$ (therefore $\delta$) and priors.

**Corollary 3** (Downsampling). *The solution to the downsampled problem is given by taking $\tilde{\delta} \triangleq \frac{\delta}{2\pi_+}$ for $\delta$ in Theorem 1 and setting $\pi_+ = \pi_- = \frac{1}{2}$ and $\rho = 1$. For square loss, the closed form follows from Corollary 2.*

We see an illustration of the relative strengths of these methods in Figure 5 which shows that the increased effective overparameterization for downsampling results in much higher error for both classes. Note that as $\delta$ approaches 0, the two methods perform equally. This aligns with the theory of cost-sensitive population risk minimization [3]. The increasing gap between wERM and downsampling is indicative of the strength of wERM as overparameterization increases. This is particularly important for highly imbalanced datasets as the effective overparameterization used in downsampling grows as $\frac{1}{\pi_+}$.

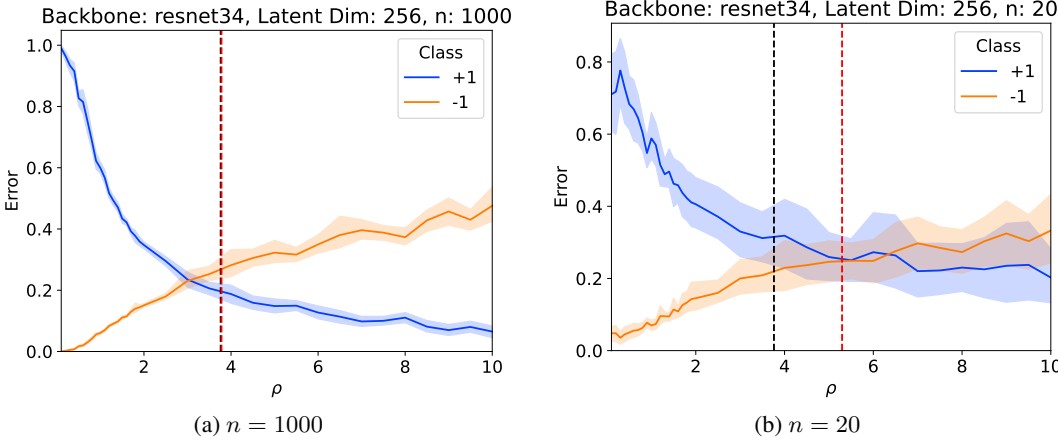

(a) $n = 1000$          (b) $n = 20$

Figure 6: Per-class accuracies for binary hair texture classification on the CelebA dataset, with the classical ratio of the priors $\rho$ marked in black and our $\tilde{\rho}$ in red. We see that the number of samples (thus $\delta$) affects the empirically optimal weighting, with larger $\delta$ requiring a larger weight on the minority class. This aligns with the theoretical results of Theorem 2.

## 4 Application to Imbalanced Image Classification

We have presented a theoretical framework for understanding how overparameterization affects the cost-sensitive learning methodologies, and we next explore this effect in practice for binary image classification tasks. Additional plots can be found in Appendix B. All relevant hyperparameters are discussed in Appendix C.

We consider the CelebA [25] dataset which consists of images of celebrity faces, each marked with 40 binary attributes. We select `Straight Hair` as the attribute of interest with takes the value 1 in 21% of samples and the value 0 in 79% of samples. We also consider a binary version of CIFAR10 [26] with artificial imbalance, where class $-1$ is `truck` with 91% of the data and $+1$ is `airplane` with 9% of the data. We consider other classes and imbalance ratios within CIFAR10 in Appendix B and find qualitatively similar behavior.

We finetune a ResNet34 model on the training split of each dataset using cross entropy loss before retraining the final layer with varying $\rho$ on the validation split with square loss (aligning with theory). The focus on square loss may seem like a major restriction, but in practice, square loss is often as performant as cross entropy loss, especially in the low-data fine-tuning setting [27, 28]. To simulate different $\delta$ values, since the model size is fixed for LLR, we subsample the validation data uniformly to size $n$. For each $n$, this retraining is repeated 10 times with different subsamples to get confidence intervals on the captured metrics. We note that CIFAR10 does not provide a validation split, so we create one from fixed 10% split of the test data. For this reason, we only consider up to $n = 90$ for CIFAR10.

We note that while we choose the size of the latent space, this is not the $d$ that is used for calculating $\delta$. We observe in practice that the majority of the features in the latent space are irrelevant (see Figure 9 in Appendix B for PCA spectra). As such, we perform PCA to quantify the number of "effective" dimensions in the data as the number of features which capture 99% of the variance. This is 3 dimensions for CelebA and 2 for CIFAR10. In general, the number of effective dimensions will be a function of task difficulty and model architecture. The relatively poor usage of the latent space is a core observation of Kirichenko et al. [1] and motivates sparse LLR. Here, we quantify the level of sparsity to better select importance weights.

We see in Figure 6 that while the $\tilde{\rho}$ calculated using the effective dimension does not perfectly predict the optimal empirical $\rho$, it captures the correct behavior as $n$ shrinks (and therefore $\delta$ increases). Note that for $n = 1000$, the population-optimal weights are not empirically optimal, likely due to a small shift from the validation data to the test data or inadequate model generalization. For $n = 20$, $\tilde{\rho}$ is distinctly larger than the ratio of the priors and achieves much better WCE.

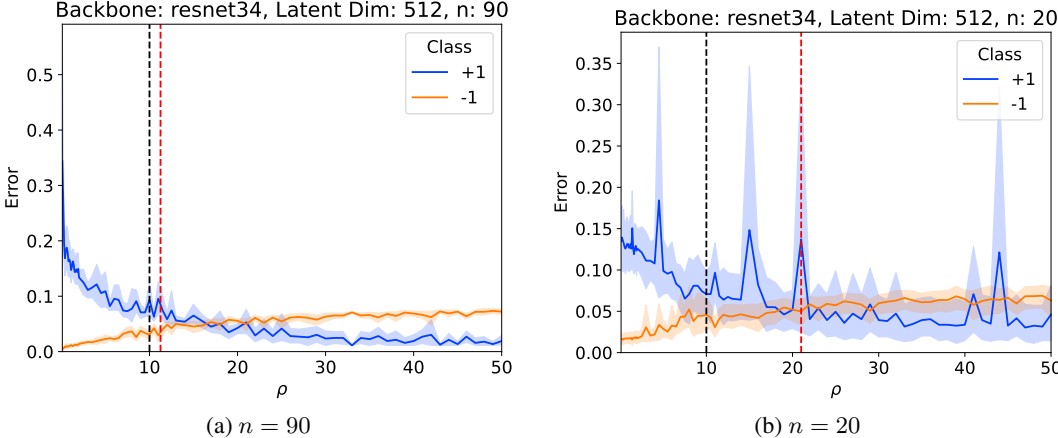

(a) $n = 90$            (b) $n = 20$

Figure 7: Per-class accuracies for binary classification on the CIFAR10 dataset (planes vs trucks), with the classical ratio of the priors $\rho$ marked in black and our $\tilde{\rho}$ in red. We see that the number of samples (thus $\delta$) affects the empirically optimal weighting, with larger $\delta$ requiring a larger weight on the minority class. This aligns with the theoretical results of Theorem 2.

In Figure 7, for the CIFAR10 dataset, we see that $\tilde{\rho}$ calculated from the effective latent dimension (red) predicts the empirical optimal $\rho$ quite well, and is certainly better in terms of WCE than the classical ratio of the priors (black). While there is significant noise with only $n = 20$ retraining samples (so only 1 or 2 minority examples), weighting with $\tilde{\rho}$ allows us to recover a more balanced classifier.

Ultimately, we see that the optimal weighting scheme is effective across different datasets and over-parameterization levels, including other choices of class for CIFAR10 (see Appendix B). This suggests that our method is quite general and could be useful in practice, acting as a default weighting rather than the classical ratio of the priors.

## 5   Limitations and Broader Impacts

**Limitations**   While it is common to assume that the latent space of a deep model is Gaussian, a more sophisticated mixture could be useful in explaining class-imbalanced learning under spurious correlation or related settings. Our analysis is also limited to isotropic noise, but we hope to extend this in future works. Despite these assumptions in the analysis, we see that in real datasets, where neither of these hold, the derived $\tilde{\rho}$ can still be useful in selecting an appropriate weighting during last layer retraining.

While our provided analysis is able to effectively capture the behavior of real models under last layer retraining with square loss, providing a clear prescription for weighting depends on the "effective" dimension of the latent data. We utilize PCA, but this is rather heuristic. A weighting methodology which learns this effective dimension in a more principled manner could increase the practical application of our findings and is a major focus of our future work.

**Multiclass Classification**   While we limit our study to binary classification for the theoretical contributions, there has been recent work [29, 30] exploring similar technical tools in the multiclass setting. This is beyond the technical scope of our work, but is a promising direction for future work. We conjecture that the intuition developed with our theory will prove useful even in the more complex multiclass setting. Specifically, we expect that for small $n$, the weights needed to reach balanced accuracy will be more extreme (favoring the smallest class) than the naive ratio of priors suggests.

**Broader impacts**   There is ongoing discussion about whether improving the performance of minority classes at the expense of majority classes is desirable. However, in critical applications such as disease or rare event detection, our weighting scheme could result in significant gains on rare classes which often have an outsized real-world impact.

# 6   Conclusion

In this work, we have demonstrated the efficacy of loss weighting in the last layer retraining setting and derived an optimal weighting scheme which takes into account overparameterization. Our results close a wide gap in the literature between the population (many samples) setting and the overparameterized (few samples) setting and provide new insights into this highly relevant regime.

In practice, we show that this novel weighting scheme outperforms the classical ratio of priors in vision tasks, but that a different notion of dimension needs to be taken into account. This compensates for the fact that many dimensions of the last layer of large models go unutilized, a core observation of previous works. Our prescription for loss weighting is general in that it outperforms the ratio of the priors across datasets, latent dimensions, and imbalance ratios in real-world settings. Our findings not only challenge the assumption that the classical ratio of priors is optimal but also provides a pathway towards more balanced model retraining by leveraging this effective dimension. We believe these insights will extend the benefits of optimal loss weighting to more complex real-world fine-tuning applications.

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

# A  Proofs

## A.1  Proof of Risk in (1)

*Proof.* Note we can decompose $\boldsymbol{X}$ as

$$\underline{X} = y\underline{\mu} + \underline{Z}, \quad \underline{Z} \sim \mathcal{N}(0, \mathbf{I}_d) \tag{13}$$

or in matrix notation as

$$\boldsymbol{X} = \underline{y}\mu^\top + \boldsymbol{Z}, \quad \underline{Z}_i \sim \mathcal{N}(0, \mathbf{I}_d), i \in [n] \tag{14}$$

Then

$$\mathcal{R}_+ = \Pr(Y(\boldsymbol{X}^\top \underline{\theta} + b) < 0 | Y = 1) \tag{15}$$

$$= \Pr(\mu^\top \underline{\theta} + \underline{Z}^\top \underline{\theta} + b < 0) \tag{16}$$

$$= \Pr(\underline{Z}^\top \underline{\theta} > \mu^\top \underline{\theta} + b) \tag{17}$$

$$= \Pr(\|\underline{\theta}\| Z' > \mu^\top \underline{\theta} + b) \tag{18}$$

$$= \Pr(Z' > \frac{\gamma s}{\alpha} + \frac{b}{\alpha}) \tag{19}$$

$$= Q(\frac{\gamma s}{\alpha} + \frac{b}{\alpha}) \tag{20}$$

where $\underline{Z} \sim \mathcal{N}(\underline{0}, \boldsymbol{I})$ and $Z' \sim \mathcal{N}(0, 1)$. $\square$

## A.2  Technical Tool: CGMT

A key technical tool in this work is the convex Gaussian minimax theorem, and its predecessor Gordon's comparison lemma. We restate the key result here, but refer the reader to Thrampoulidis et al. [23] for a more detailed view on the problem.

Consider a pair of primary and auxiliary optimization problems:

$$\Phi(\boldsymbol{G}) \triangleq \min_{\underline{w} \in \mathcal{S}_w} \max_{\underline{u} \in \mathcal{S}_u} \underline{u}^\top \boldsymbol{G} \underline{w} + \psi(\underline{w}, \underline{u}), \tag{21}$$

$$\phi(\underline{g}, \underline{h}) \triangleq \max_{\underline{w} \in \mathcal{S}_w} \min_{\underline{u} \in \mathcal{S}_u} \|\underline{w}\|_2 \underline{g}^\top \underline{u} + \|\underline{u}\|_2 \underline{h}^\top \underline{w} + \psi(\underline{w}, \underline{u}). \tag{22}$$

We will show that for certain random inputs, these two problems are equivalent.

**Theorem 4** (CGMT [22]). *Let* $\Phi, \phi$ *be defined as in* (21) *and* (22), $\boldsymbol{G}$ *is a matrix with standard Gaussian entries, and* $\underline{G}, \underline{H} \sim \mathcal{N}(0, \mathbf{I})$.

$$\Pr\left(\Phi(\boldsymbol{G}) < c\right) \leq 2 \Pr\left(\phi(\underline{G}, \underline{H}) \leq c\right) \tag{23}$$

*additionally if* $\psi$ *is convex-concave, then* $\forall c \in \mathbb{R}$

$$\Pr\left(\Phi(\boldsymbol{G}) > c\right) \leq 2 \Pr\left(\phi(\underline{G}, \underline{H}) \geq c\right). \tag{24}$$

*That is, we can bound the performance of* $\Phi$ *by the comparatively simpler* $\phi$.

We will leverage this result to study the comparatively simple auxiliary optimization problem through a series of scalarizations.

## A.3  Properties of the Moreau Envelope

**Lemma 1** (Properties of Moreau). *Let* $\ell : \mathbb{R} \to \mathbb{R}$ *be proper, closed, convex function. The Moreau envelope* $\mathcal{M}_\ell(x; \lambda)$ *is jointly convex in its arguments and the following hold:*

$$\mathcal{M}'_{\ell,1}(x; \lambda) := \frac{x - \text{prox}_\ell(x; \lambda)}{\lambda}, \tag{25}$$

$$\mathcal{M}'_{\ell,2}(x; \lambda) := -\frac{1}{2} \left(\mathcal{M}'_{\ell,1}(x; \lambda)\right)^2. \tag{26}$$

*For a complete list of useful properties see for example Lemma D.1 in [23].*

## A.4 Proof of Theorem 1

Note that we can rewrite (4) by the constrained optimization problem

$$\min_{\underline{\theta}, b} \quad \frac{1}{n} \sum_{i=1}^{n} \omega_i \ell(u_i) \tag{27}$$

$$\text{s.t.} \quad u_i = y_i(\underline{x}_i^\top \underline{\theta} + b) \quad \forall i \in [n]$$

which then allows us to write the problem as a min-max:

$$\min_{\underline{\theta}, b, \underline{u}} \max_{\underline{v}} \frac{1}{n} \sum_{i=1}^{n} v_i(u_i - y_i \underline{x}_i^\top \underline{\theta} - y_i b) + \omega_i \ell(u_i). \tag{28}$$

From this point, we can move to matrix notation

$$\min_{\underline{\theta}, b, \underline{u}} \max_{\underline{v}} \frac{1}{n} \underline{v}^\top (\underline{u} - \boldsymbol{D}_y \boldsymbol{X} \underline{\theta} - \underline{y} b) + \mathcal{L}_\omega(\underline{u}), \tag{29}$$

where $\mathcal{L}_\omega(\underline{u}) \triangleq \frac{1}{n} \sum_{i=1}^{n} \omega_i \ell(u_i)$ and $\boldsymbol{D}_y$ is the diagonal matrix of the vector $\underline{y}$. We can further decompose this by expanding $\boldsymbol{X}$ as

$$\min_{\underline{\theta}, b, \underline{u}} \max_{\underline{v}} \frac{1}{n} [\underline{v}^\top \underline{u} - \underline{v}^\top \boldsymbol{D}_y (\underline{y} \mu^\top - \boldsymbol{Z}) \underline{\theta} - \underline{v}^\top \underline{y} b] + \mathcal{L}_\omega(\underline{u}). \tag{30}$$

Finally, noting that $\boldsymbol{D}_y \underline{y} = \underline{1}$, we have the primary optimization:

$$\min_{\underline{\theta}, b, \underline{u}} \max_{\underline{v}} \frac{1}{n} [\underline{v}^\top \boldsymbol{Z} \underline{\theta} - \underline{v}^\top \underline{1} (\mu^\top \underline{\theta}) - \underline{v}^\top \underline{y} b + \underline{v}^\top \underline{u}] + \mathcal{L}_\omega(\underline{u}). \tag{PO}$$

Next we write our auxiliary optimization following CGMT Theorem 4

$$\min_{\underline{\theta}, b, \underline{u}} \max_{\underline{v}} \frac{1}{n} [\|\underline{\theta}\| \underline{g}^\top \underline{v} + \|\underline{v}\| \underline{h}^\top \underline{\theta} + \underline{v}^\top \underline{u} - \underline{v}^\top \underline{1} (\mu^\top \underline{\theta}) - \underline{v}^\top \underline{y} b] + \mathcal{L}_\omega(\underline{u}). \tag{AO}$$

We can separately optimize over the direction and magnitude of $\underline{v}$:

$$\min_{\underline{\theta}, b, \underline{u}} \max_{\beta \geq 0, \|\underline{v}\| = \beta} \frac{1}{n} [\|\underline{\theta}\| \underline{g}^\top \underline{v} + \beta \underline{h}^\top \underline{\theta} + \underline{v}^\top \underline{u} - \underline{v}^\top \underline{1} (\mu^\top \underline{\theta}) - \underline{v}^\top \underline{y} b] + \mathcal{L}_\omega(\underline{u}) \tag{31}$$

$$\min_{\underline{\theta}, b, \underline{u}} \max_{\beta \geq 0, \|\underline{v}\| = \beta} \frac{1}{n} [\underline{v}^\top (\|\underline{\theta}\| \underline{g} + \underline{u} - \underline{1} (\mu^\top \underline{\theta}) - \underline{y} b) + \beta \underline{h}^\top \underline{\theta}] + \mathcal{L}_\omega(\underline{u}) \tag{32}$$

$$\min_{\underline{\theta}, b, \underline{u}} \max_{\beta \geq 0} \frac{1}{n} [\beta \|\underline{u} + \|\underline{\theta}\| \underline{g} - \underline{1} (\mu^\top \underline{\theta}) - \underline{y} b\| + \beta \underline{h}^\top \underline{\theta}] + \mathcal{L}_\omega(\underline{u}). \tag{33}$$

Next we use the AM-GM trick[1] to rewrite the two norm as the squared two norm, letting $\xi = \frac{\beta}{\sqrt{n}}$:

$$\min_{\underline{\theta}, b, \underline{u}} \max_{\xi \geq 0} \min_{\tau > 0} \frac{\xi \tau}{2} + \frac{\xi}{2 \tau n} \|\underline{u} + \|\underline{\theta}\| \underline{g} - \underline{1} (\mu^\top \underline{\theta}) - \underline{y} b\|^2 + \frac{\xi}{\sqrt{n}} \underline{h}^\top \underline{\theta} + \mathcal{L}_\omega(\underline{u}) \tag{34}$$

Now we optimize $\underline{\theta}$ in the orthogonal subspace to $\mu$ setting $\|\underline{\theta}\| = \alpha$ and $\frac{\underline{\theta}^\top \mu}{\|\mu\|} = \gamma$ [2]:

$$\min_{\alpha, b, \gamma \leq \alpha, u} \max_{\xi \geq 0} \min_{\tau > 0} \frac{\xi \tau}{2} + \frac{\xi}{2 \tau n} \|\underline{u} + \alpha \underline{g} - \underline{1} (\|\mu\| \gamma) - \underline{y} b\|^2$$
$$- \frac{\xi}{\sqrt{n}} (\underline{h}^\top \mu \frac{\gamma}{\|\mu\|} + \sqrt{\alpha^2 - \gamma^2} \|\boldsymbol{P}^\perp \underline{h}\|) + \mathcal{L}_\omega(\underline{u}) \tag{35}$$

---

[1] $\frac{x}{\sqrt{n}} = \min_{\tau > 0} \frac{\tau}{2} + \frac{x^2}{2\tau}$

[2] $\underline{\theta} = \frac{(\mu^\top \underline{\theta})}{\|\mu\|} \frac{\mu}{\|\mu\|} + P^\perp \underline{\theta} \Rightarrow \|\underline{\theta}\|^2 = (\frac{(\mu^\top \underline{\theta})}{\|\mu\|})^2 + \|P^\perp \underline{\theta}\|^2 \Rightarrow \|P^\perp \underline{\theta}\| = \sqrt{\alpha^2 - \gamma^2}$

where $\boldsymbol{P}^\perp$ projection into the orthogonal subspace of $\underline{\mu}$. This can then be written in terms of the Moreau envelope:

$$\min_{\alpha,b,\gamma\leq\alpha}\max_{\xi\geq0}\min_{\tau>0}\frac{\xi\tau}{2}-\frac{\xi}{\sqrt{n}}(\underline{h}^\top\underline{\mu}\frac{\gamma}{\|\underline{\mu}\|}+\sqrt{\alpha^2-\gamma^2}\|\boldsymbol{P}^\perp\underline{h}\|)$$
$$+\frac{1}{n}[\sum_{i=1}^{n_+}\omega_+\mathcal{M}(-\alpha g_i+\|\underline{\mu}\|\gamma+b;\frac{\tau\omega_+}{\xi})$$
$$+\sum_{j=1}^{n_-}\omega_-\mathcal{M}(-\alpha g_j+\|\underline{\mu}\|\gamma-b;\frac{\tau\omega_-}{\xi})]. \tag{36}$$

where we have grouped by $\underline{y}$. Letting $n,d\to\infty$ with $\frac{d}{n}=\delta$, we have

$$\min_{\alpha>0,b,\gamma\leq\alpha,\tau>0}\max_{\xi\geq0}\frac{\xi\tau}{2}-\xi\sqrt{\delta(\alpha^2-\gamma^2)}+\pi_+\omega_+\mathbb{E}[\mathcal{M}(-\alpha G+\|\underline{\mu}\|\gamma+b;\frac{\tau\omega_+}{\xi})]$$
$$+\pi_-\omega_-\mathbb{E}[\mathcal{M}(-\alpha G+\|\underline{\mu}\|\gamma-b;\frac{\tau\omega_-}{\xi})]. \tag{37}$$

where we have switched the order of the min and max. See Taheri et al. [31] for a justification and further technical details.

This leaves us with a simplified systems of equations to solve based on first-order optimality conditions:

$$\frac{\xi}{2}+\pi_+\frac{\omega_+^2}{\xi}\mathbb{E}[\mathcal{M}'_{\ell,2}(-\alpha G+\|\underline{\mu}\|\gamma+b;\frac{\tau\omega_+}{\xi})]$$
$$+\pi_-\frac{\omega_-^2}{\xi}\mathbb{E}[\mathcal{M}'_{\ell,2}(-\alpha G+\|\underline{\mu}\|\gamma-b;\frac{\tau\omega_-}{\xi})]=0 \tag{38a}$$

$$\frac{\tau}{2}-\sqrt{\delta(\alpha^2-\gamma^2)}-\pi_+\frac{\tau\omega_+^2}{\xi^2}\mathbb{E}[\mathcal{M}'_{\ell,2}(-\alpha G+\|\underline{\mu}\|\gamma+b;\frac{\tau\omega_+}{\xi})]$$
$$-\pi_-\frac{\tau\omega_-^2}{\xi^2}\mathbb{E}[\mathcal{M}'_{\ell,2}(-\alpha G+\|\underline{\mu}\|\gamma-b;\frac{\tau\omega_-}{\xi})]=0 \tag{38b}$$

$$\xi\frac{\delta\gamma}{\sqrt{\delta(\alpha^2-\gamma^2)}}+\pi_+\omega_+\|\underline{\mu}\|\mathbb{E}[\mathcal{M}'_{\ell,1}(-\alpha G+\|\underline{\mu}\|\gamma+b;\frac{\tau\omega_+}{\xi})]$$
$$+\pi_-\omega_-\|\underline{\mu}\|\mathbb{E}[\mathcal{M}'_{\ell,1}(-\alpha G+\|\underline{\mu}\|\gamma-b;\frac{\tau\omega_-}{\xi})]=0 \tag{38c}$$

$$\pi_+\omega_+\mathbb{E}[\mathcal{M}'_{\ell,1}(-\alpha G+\|\underline{\mu}\|\gamma+b;\frac{\tau\omega_+}{\xi})]$$
$$-\pi_-\omega_-\mathbb{E}[\mathcal{M}'_{\ell,1}(-\alpha G+\|\underline{\mu}\|\gamma-b;\frac{\tau\omega_-}{\xi})]=0 \tag{38d}$$

$$-\xi\frac{\delta\alpha}{\sqrt{\delta(\alpha^2-\gamma^2)}}-\pi_+\omega_+\mathbb{E}[G\mathcal{M}'_{\ell,1}(-\alpha G+\|\underline{\mu}\|\gamma+b;\frac{\tau\omega_+}{\xi})]$$
$$-\pi_-\omega_-\mathbb{E}[G\mathcal{M}'_{\ell,1}(-\alpha G+\|\underline{\mu}\|\gamma-b;\frac{\tau\omega_-}{\xi})]=0 \tag{38e}$$

Let $\lambda'\triangleq\tau/\xi$. Combining (38a) and (38b) we obtain $\tau=\sqrt{\delta(\alpha^2-\gamma^2)}$, equivalently, we have $\xi=\sqrt{\delta(\alpha^2-\gamma^2)}/\lambda'$. Substituting this into (38c) and (38e) we see the following simplifications:

$$\delta(\alpha^2-\gamma^2)+2\lambda'^2\pi_+\omega_+^2\mathbb{E}[\mathcal{M}'_{\ell,2}(-\alpha G+\|\underline{\mu}\|\gamma+b;\lambda'\omega_+)]$$
$$+2\lambda'^2\pi_-\omega_-^2\mathbb{E}[\mathcal{M}'_{\ell,2}(-\alpha G+\|\underline{\mu}\|\gamma-b;\lambda'\omega_-)]=0 \tag{39a}$$

$$\frac{\delta\gamma}{\lambda'}+\pi_+\omega_+\|\underline{\mu}\|\mathbb{E}[\mathcal{M}'_{\ell,1}(-\alpha G+\|\underline{\mu}\|\gamma+b;\lambda'\omega_+)]$$
$$+\pi_-\omega_-\|\underline{\mu}\|\mathbb{E}[\mathcal{M}'_{\ell,1}(-\alpha G+\|\underline{\mu}\|\gamma-b;\lambda'\omega_-)]=0 \tag{39b}$$

$$-\frac{\delta\alpha}{\lambda'}-\pi_+\omega_+\mathbb{E}[G\mathcal{M}'_{\ell,1}(-\alpha G+\|\underline{\mu}\|\gamma+b;\lambda'\omega_+)]$$

$$-\pi_-\omega_-\mathbb{E}[G\mathcal{M}'_{\ell,1}(-\alpha G + \|\underline{\mu}\|\gamma - b; \lambda'\omega_-)] = 0 \tag{39c}$$

$$\pi_+\omega_+\mathbb{E}[\mathcal{M}'_{\ell,1}(-\alpha G + \|\underline{\mu}\|\gamma + b; \lambda'\omega_+)]$$
$$-\pi_-\omega_-\mathbb{E}[\mathcal{M}'_{\ell,1}(-\alpha G + \|\underline{\mu}\|\gamma - b; \lambda'\omega_-)] = 0 \tag{39d}$$

Note that the above depends only on the ratio of the weights, so let $\lambda \triangleq \lambda'\omega_+$ and $\rho \triangleq \frac{\omega_+}{\omega_-}$,

$$\delta(\alpha^2 - \gamma^2) + 2\lambda^2\pi_+\mathbb{E}[\mathcal{M}'_{\ell,2}(-\alpha G + \|\underline{\mu}\|\gamma + b; \lambda)]$$
$$+2\frac{\lambda^2}{\rho^2}\pi_-\mathbb{E}[\mathcal{M}'_{\ell,2}(-\alpha G + \|\underline{\mu}\|\gamma - b; \lambda/\rho)] = 0 \tag{40a}$$

$$\frac{\delta\gamma\rho}{\lambda} + \pi_+\rho\|\underline{\mu}\|\mathbb{E}[\mathcal{M}'_{\ell,1}(-\alpha G + \|\underline{\mu}\|\gamma + b; \lambda)]$$
$$+\pi_-\|\underline{\mu}\|\mathbb{E}[\mathcal{M}'_{\ell,1}(-\alpha G + \|\underline{\mu}\|\gamma - b; \lambda/\rho)] = 0 \tag{40b}$$

$$-\frac{\delta\alpha\rho}{\lambda} - \pi_+\rho\mathbb{E}[G\mathcal{M}'_{\ell,1}(-\alpha G + \|\underline{\mu}\|\gamma + b; \lambda)]$$
$$-\pi_-\mathbb{E}[G\mathcal{M}'_{\ell,1}(-\alpha G + \|\underline{\mu}\|\gamma - b; \lambda/\rho)] = 0 \tag{40c}$$

$$\pi_+\rho\mathbb{E}[\mathcal{M}'_{\ell,1}(-\alpha G + \|\underline{\mu}\|\gamma + b; \lambda)]$$
$$-\pi_-\mathbb{E}[\mathcal{M}'_{\ell,1}(-\alpha G + \|\underline{\mu}\|\gamma - b; \lambda/\rho)] = 0 \tag{40d}$$

We can rewrite using Stein's lemma[3]:

$$\delta(\alpha^2 - \gamma^2) + 2\lambda^2\pi_+\mathbb{E}[\mathcal{M}'_{\ell,2}(-\alpha G + \|\underline{\mu}\|\gamma + b; \lambda)]$$
$$+2\lambda^2/\rho^2\pi_-\mathbb{E}[\mathcal{M}'_{\ell,2}(-\alpha G + \|\underline{\mu}\|\gamma - b; \lambda/\rho)] = 0 \tag{41a}$$

$$\frac{\delta\gamma\rho}{\lambda} + \pi_+\rho\|\underline{\mu}\|\mathbb{E}[\mathcal{M}'_{\ell,1}(-\alpha G + \|\underline{\mu}\|\gamma + b; \lambda)]$$
$$+\pi_-\|\underline{\mu}\|\mathbb{E}[\mathcal{M}'_{\ell,1}(-\alpha G + \|\underline{\mu}\|\gamma - b; \lambda/\rho)] = 0 \tag{41b}$$

$$-\frac{\delta\alpha\rho}{\lambda} + \pi_+\rho\alpha\mathbb{E}[\mathcal{M}''_{\ell,1}(-\alpha G + \|\underline{\mu}\|\gamma + b; \lambda)]$$
$$+\pi_-\alpha\mathbb{E}[\mathcal{M}''_{\ell,1}(-\alpha G + \|\underline{\mu}\|\gamma - b; \lambda/\rho)] = 0 \tag{41c}$$

$$\pi_+\rho\mathbb{E}[\mathcal{M}'_{\ell,1}(-\alpha G + \|\underline{\mu}\|\gamma + b; \lambda)]$$
$$-\pi_-\mathbb{E}[\mathcal{M}'_{\ell,1}(-\alpha G + \|\underline{\mu}\|\gamma - b; \lambda/\rho)] = 0 \tag{41d}$$

This completes the proof.

### A.5  Proof of Corollary 1

The margin-based form of square loss can be written as

$$\ell_{\text{square}}(z) \triangleq \frac{1}{2}(z - 1)^2 \tag{42}$$

where $z$ is the margin. The prox operator (and therefore the Moreau envelope) has a nice form for this loss:

$$\text{prox}_{\ell_{\text{square}}}(x; \lambda) = \frac{x + \lambda}{1 + \lambda} \tag{43}$$

which thus reduces the Moreau envelope to

$$\mathcal{M}_{\ell_{\text{square}}}(x; \lambda) = \frac{1}{2}(\frac{x + \lambda}{1 + \lambda} - 1)^2 + \frac{1}{2\lambda}(\frac{\lambda(1 - x)}{1 + \lambda})^2 \tag{44a}$$

$$= \frac{1}{2}\frac{(x - 1)^2}{(1 + \lambda)^2} + \frac{\lambda(1 - x)^2}{2(1 + \lambda)^2} \tag{44b}$$

$$= \frac{1}{2}\frac{(x - 1)^2}{1 + \lambda} \tag{44c}$$

---

[3]For smooth function $f$ and $G$ standard normal: $\mathbb{E}[Gf(G)] = \mathbb{E}[f'(G)]$ and $\mathbb{E}[Gf(aG)] = a\mathbb{E}[f'(aG)]$

We have the following derivatives of the Moreau envelope:

$$\mathcal{M}'_{\ell,1}(x;\lambda) := \frac{x-1}{1+\lambda} \tag{45}$$

$$\mathcal{M}''_{\ell,1}(x;\lambda) := \frac{1}{1+\lambda} \tag{46}$$

$$\mathcal{M}'_{\ell,2}(x;\lambda) := \frac{-(x-1)^2}{2(1+\lambda)^2} \tag{47}$$

Using the properties of the Moreau envelope of square loss, we get the following from (41):

$$\delta(\alpha^2 - \gamma^2) + \lambda^2 \pi_+ \frac{-\mathbb{E}[(-\alpha G + \|\underline{\mu}\|\gamma + b - 1)^2]}{(1+\lambda)^2}$$
$$+ \lambda^2/\rho^2 \pi_- \frac{-\mathbb{E}[(-\alpha G + \|\underline{\mu}\|\gamma - b - 1)^2]}{(1+\lambda/\rho)^2} = 0 \tag{48a}$$

$$\frac{\delta\gamma\rho}{\lambda} + \pi_+ \rho\|\underline{\mu}\| \frac{\mathbb{E}[(-\alpha G + \|\underline{\mu}\|\gamma + b - 1)]}{1+\lambda}$$
$$+ \pi_-\|\underline{\mu}\| \frac{\mathbb{E}[(-\alpha G + \|\underline{\mu}\|\gamma - b - 1)]}{1+\lambda/\rho} = 0 \tag{48b}$$

$$-\frac{\delta\alpha\rho}{\lambda} + \pi_+ \alpha\rho \frac{1}{1+\lambda} + \pi_- \alpha \frac{1}{1+\lambda/\rho} = 0 \tag{48c}$$

$$\pi_+ \rho \frac{\mathbb{E}[(-\alpha G + \|\underline{\mu}\|\gamma + b - 1)]}{1+\lambda}$$
$$- \pi_- \frac{\mathbb{E}[(-\alpha G + \|\underline{\mu}\|\gamma - b - 1)]}{1+\lambda/\rho} = 0 \tag{48d}$$

Taking the expectations, we get

$$\delta(\alpha^2 - \gamma^2) - \lambda^2 \pi_+ \frac{2b\gamma\|\underline{\mu}\| + \gamma^2\|\underline{\mu}\|^2 - 2\gamma\|\underline{\mu}\| + b^2 - 2b + \alpha^2 + 1}{(1+\lambda)^2}$$
$$- \lambda^2/\rho^2 \pi_- \frac{-2b\gamma\|\underline{\mu}\| + \gamma^2\|\underline{\mu}\|^2 - 2\gamma\|\underline{\mu}\| + b^2 + 2b + \alpha^2 + 1}{(1+\lambda/\rho)^2} = 0 \tag{49a}$$

$$\frac{\delta\gamma\rho}{\lambda} + \pi_+ \rho\|\underline{\mu}\| \frac{\|\underline{\mu}\|\gamma + b - 1}{1+\lambda} + \pi_-\|\underline{\mu}\| \frac{\|\underline{\mu}\|\gamma - b - 1}{1+\lambda/\rho} = 0 \tag{49b}$$

$$-\frac{\delta\rho}{\lambda} + \pi_+ \rho \frac{1}{1+\lambda} + \pi_- \frac{1}{1+\lambda/\rho} = 0 \tag{49c}$$

$$\pi_+ \rho \frac{\|\underline{\mu}\|\gamma + b - 1}{1+\lambda} - \pi_- \frac{\|\underline{\mu}\|\gamma - b - 1}{1+\lambda/\rho} = 0 \tag{49d}$$

Further, we can examine the sum of the weighted expected Moreau envelopes and see by direct calculation that the sum is jointly strictly convex in $\alpha, \gamma, b, \lambda$. Of note, each expected Moreau envelope on its own is not strictly convex (in fact the determinant of the hessian is 0). This proves Corollary 1.

## A.6 Comparison of $\rho = 1$ and $\tilde{\rho}$

Note that we have closed forms for both the unweighted case (Corollary 2) and the weighted case such that $b^* = 0$ (Theorem 2). Thus, it is natural to compare the two to see when it is advantageous to weight in this manner.

In the unweighted case, the WCE is dominated by the positive (minority) risk in (1) since $b^* < 0$. On the other hand, the two class-conditional risks are equal for the model learned with $\tilde{\rho}$ by construction. Thus, we need not consider the max in the definition of WCE (2). The comparison of interest, then, is simply two $Q$ functions. We will derive conditions under which $\tilde{\rho}$ achieves lower WCE than

$\rho = 1$. We denote the solution to the weighted problem as $(\tilde{\gamma}, \tilde{\alpha})$ and the unweighted problem as $(\gamma^*, \alpha^*, b^*)$. Thus, in order for the $\tilde{\rho}$-reweighted WCE to be lower than the unweighted WCE,

$$Q\left(\frac{\tilde{\gamma}s}{\tilde{\alpha}}\right) \leq Q\left(\frac{\gamma^*s + b^*}{\alpha^*}\right),$$ (50)

requiring

$$\frac{\tilde{\gamma}s}{\tilde{\alpha}} \geq \frac{\gamma^*s + b^*}{\alpha^*}.$$ (51)

Substituting the closed-form solutions yields the following necessary and sufficient condition:

$$s^2 \leq \frac{1 - 2\pi_+}{2\left(2\pi_+(1 - \pi_+) - \sqrt{\frac{(1-\pi_+)\pi_+(4(1-\pi_+)\pi_+ - \delta)}{1-\delta}}\right)}$$ (52)

This final expression demonstrates that if the separation of the classes is too large, then the unweighted model will outperform the weighted model. This is perhaps intuitive given that large separations essentially act as leverage, possibly causing the weights to overcorrect. Note that in this large separation regime, $\frac{\tilde{\gamma}s}{\tilde{\alpha}}$ is approximately linear and so the WCE of the weighted model decays exponentially in $s$. This tells us that when $\tilde{\rho}$ is outperformed by $\rho = 1$, the errors are very small.

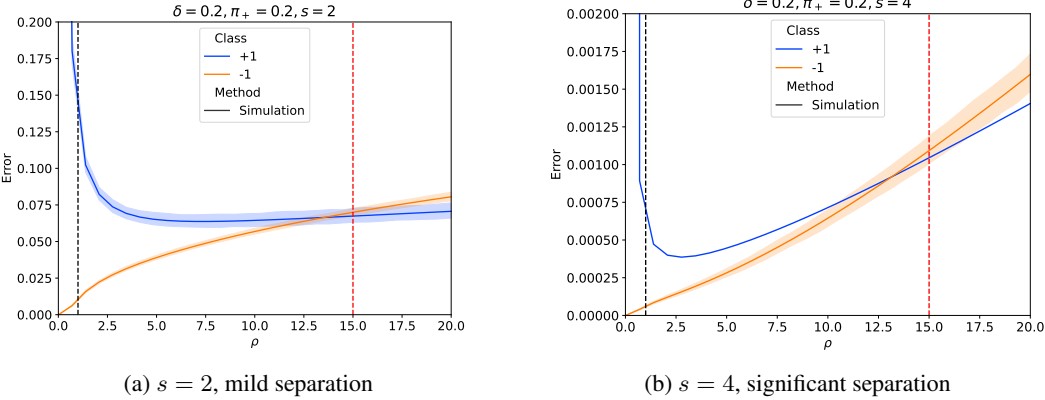

(a) $s = 2$, mild separation

(b) $s = 4$, significant separation

Figure 8: With mild separation, the weighted model (red) does significantly better than the unweighted model (black), but this is reversed with large separation between classes. Note the difference in scales of the error.

See Figure 8 to understand the behavior of the class-conditional risks with increasing separation. In Figure 8b, note that $\tilde{\rho}$ is predictive of the crossover point, but that the weight where the WCE is actually minimized is smaller than $\tilde{\rho}$ in the large separation case. This result provides clear insight into the regions where weighting is useful and emphasizes the intuitive point that larger separations demand less weighting. Indeed we see that increasing separation decreases the errors for each class while also decreasing the optimal weight; we conjecture that for $s \to \infty$, the optimal weight will decrease from $\tilde{\rho}$ to 1. This regime is of little impact, however, as the error for both will be extremely small as pointed out previously.

# B  Additional Plots

## B.1  PCA

We see in Figure 9 that very few features capture most of the variance of the retraining data. This motivates us to use the smaller "effective" dimension as mentioned in the main body. Note that for CIFAR10, the limited data limits the maximum number of meaningful features.

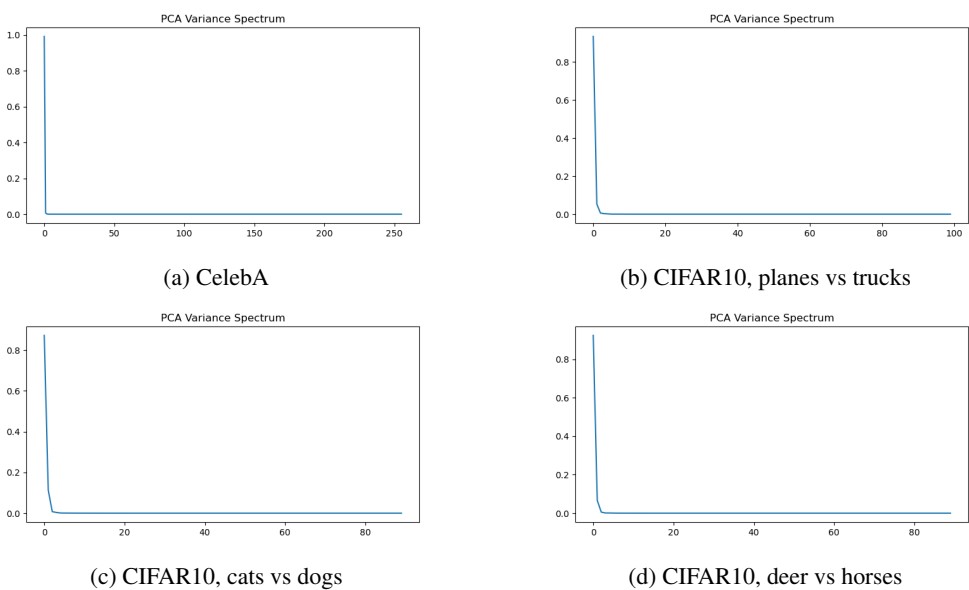

(a) CelebA

(b) CIFAR10, planes vs trucks

(c) CIFAR10, cats vs dogs

(d) CIFAR10, deer vs horses

Figure 9: PCA Spectra. We see that most of the variance is captured by very few features.

## B.2  Ablation Studies

We provide additional plots in Figure 10 showing that our findings hold even when the latent dimension of the ResNet34 model is different. Here we show results for a ResNet34 model with 128 dimensional latent space. The effective dimension (3) is still used to calculate $\tilde{\rho}$.

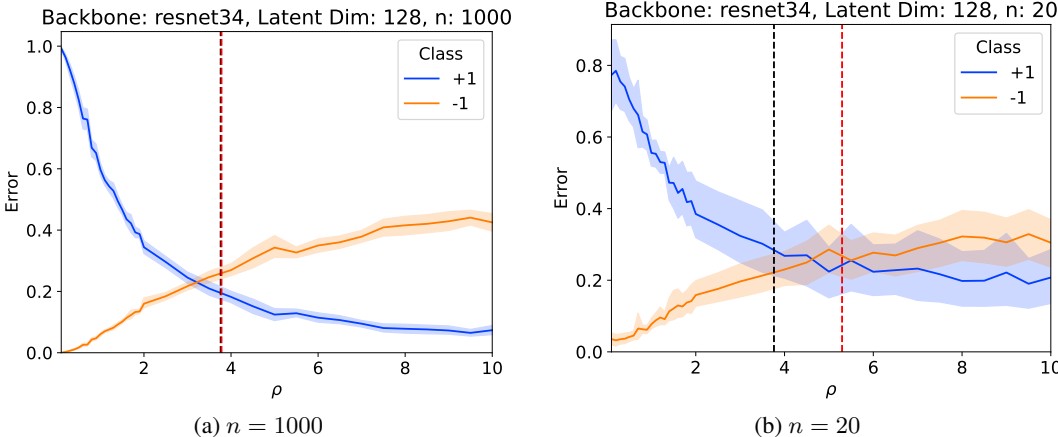

(a) $n = 1000$

(b) $n = 20$

Figure 10: Per-class errors on CelebA dataset. Even with a lower latent dimension, $\tilde{\rho}$ (red) is still predictive of the cross-over point modulo a shift seen even in the large $n$ setting. This is likely due to non-Gaussianity of the latent data.

Even for different class pairs in CIFAR10, we see similar behavior as shown in Figure 11. For this dataset, we select `cat` as the minority class $+1$ and `dog` as the majority class $-1$. The imbalance is selected as 17% class $+1$ and 83% class $-1$. For this dataset, the effective dimension is 3. $\tilde{\rho}$ is still predictive of the crossover point for both $n = 90$ and $n = 20$ resulting in significant gains over the traditional ratio of priors.

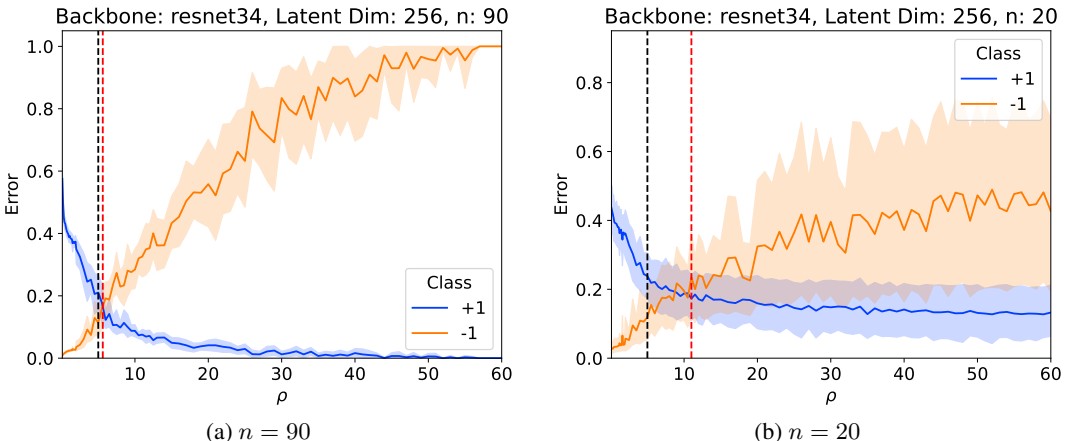

(a) $n = 90$          (b) $n = 20$

Figure 11: Per-class errors on CIFAR10 dataset, cats vs dogs. $\tilde{\rho}$ (red) is predictive of the cross-over point when using the effective dimension of the data (4).

The story is similar for `deer` vs `horse` with the same imbalance which has effective dimension 9. Note that for $n = 20$, $\tilde{\rho}$ is undefined, suggesting that the correct weighting strategy is to push $\rho \to \infty$. We see that up to $\rho = 60$, the per-class errors do not meet.

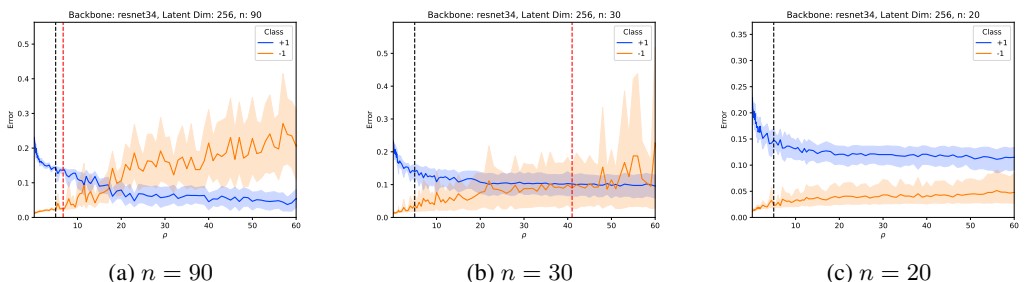

(a) $n = 90$       (b) $n = 30$       (c) $n = 20$

Figure 12: Per-class errors on CIFAR10 dataset, deer vs horses. $\tilde{\rho}$ (red) is predictive of the cross-over point when it is defined using the effective dimension of the data (9). It is undefined for $n = 20$, suggesting that the per-class risks will not meet. This effect is seen in practice.

## C   Experimental Details

All empirical experiments were performed using NVIDIA A100 GPUs while simulations were completed on CPU. Each experiment took less than 30 minutes of wall time to run after training base models. Base models took less than 4 hours to finetune from the pretrained weights. A full list of hyperparameters is provided in Table 1.

| Parameter | Value |
|---|---|
| Backbone | ResNet34 |
| Pretrained Weights | Imagenet1k-V2 |
| Latent Dimension | {128, 256, 512} |
| Optimizer | AdamW |
| Learning Rate | 1e-3 |
| Full fine-tuning epochs | 10 |
| MLP Dropout Rate | 0.5 |
| Fine-tuning epochs | 30 |
| Fine-tuning LR | 1e-2 |

Table 1: Hyperparameters for CelebA and CIFAR10

