# OpenReview forum: "Thumb on the Scale: Optimal Loss Weighting in Last Layer Retraining"
_NeurIPS.cc/2025/Conference — NeurIPS 2025 poster_

### Official Review · Reviewer_xfu4 · 2025-06-28

**Clarity:** 2
**Significance:** 2
**Originality:** 2
**Rating:** 4
**Confidence:** 1

**Summary:**

The paper investigates the class imbalance problem using Empirical Risk Minimization. The authors assume a binary classification setting and derive its closed-form solution. They then apply their method to real-world data by using subsets of the CelebA and CIFAR-10 datasets, retraining only the last layer (LLR).

**Questions:**

I do not have any further questions beyond those mentioned above.

**Ethical Concerns:**

["NO or VERY MINOR ethics concerns only"]

**Final Justification:**

Several of my concerns are resolved. However, I still have a concern regarding a multi-class scenario. I do not think this is directly applicable to that setting from the current work.

**Limitations:**

Yes

**Paper Formatting Concerns:**

I do not have any concerns regarding paper format.

**Quality:**

3

**Strengths And Weaknesses:**

[Strength]

The paper addresses classification under imbalanced data, a common issue in many natural datasets.

It is generally well written, and the accompanying code repository is clean and well-organized.


[Weakness]

 The authors assume a binary classification task and evaluate their method only on specific attributes in CelebA and a single class pair (airplanes vs. trucks) in CIFAR-10. This narrow scope limits the practical generalizability of the proposed method, even though the problem of imbalanced data is highly relevant.
The authors compare their proposed $\tilde{\rho}$ with the conventional prior ratio, but they do not explicitly define what the prior ratio is or how it is computed.

The authors need to state the terminology more carefully. For example, the $\textsf{prox}$ operator in L100 should be the proximal operator ($\textsf{prox}).  Similarly, CS-SVM in L36 and w.p. In L88, the full name should also be mentioned.

[Minor]

In L21, [1, 2] By -> [1, 2]. By

In Equation (8b), consider replacing $\delta_\gamma$ with $\delta\gamma$ if no subscript is intended.

---

> ### Author Rebuttal · Authors · 2025-07-30
>
> We thank the reviewer for their comments and suggestions. We would like to address each point presented:
>  - **W1**: We included additional results in the supplementary materials B which show similar results for additional class pairs. If accepted, we will update the manuscript with results for additional modalities such as text to expand the applicability of our work. Additionally, while our current results capture only binary classification for class imbalance, we aim to use similar techniques to tackle the more general problem of group imbalance in binary classification. We believe that this work is an important first step in understanding how the loss weighting process is affected by overparameterization.
>  - **W2**: By the conventional prior ratio, we meant the inverse ratio of the class priors, $\pi_-/\pi_+$, which is very frequently used in practice. We will ensure that this point is clarified in the next revision of this work.
>  - **W3**: We thank the reviewer for pointing out these oversights. We will ensure that any typos are fixed and the above terms are fully expanded earlier in the manuscript.

---

> > ### Comment · Reviewer_xfu4 · 2025-07-31
> >
> > I appreciate the authors for the response and thank you for pointing out Appendix B, which shows cat-dog and deer-horse pairs. However, the limited number of pairs remains a drawback. For example, the error curve for cat–dog differs markedly from that for deer–horse, indicating that additional settings should be evaluated.
> >
> > Moreover, given the extensive long-tailed classification literature on datasets such as CIFAR10-LT, CIFAR100-LT, ImageNet-LT, LVIS, and iNaturalist, restricting the theory to binary classification is a significant limitation, though I acknowledge this may be a deliberate scope choice, as authors mentioned. It would be nice if the authors made a connection between this work with multi-class classification.

---

> > > ### Author Response · Authors · 2025-08-04
> > >
> > > We thank the reviewer for their quick response and acknowledge that the an extension to the multi-class setting is an important next step to enhance the practicality of our work. As we have outlined for other reviewers, a key difficulty for the extension to the multi-class setting would be the derivation of the optimal weighting scheme. A similar scalarization is still possible with the multi-class extension of square loss, but characterizing a fair weighting scheme is significantly more difficult in the mutli-class setting without strong assumptions on the structure of the class-conditional distributions. We believe that this may be possible, and will be the focus of extensive future work.
> > >
> > > As for the results on other class pairs from CIFAR10, we note that we do not expect the curves to look exactly the same for different class pairs. In fact, we should expect that some classes may be easier or harder to distinguish from their counterpart which will undoubtedly affect the accuracy tradeoff that we see. The key observation is that the *crossover point* (the point at which the classifier achieves balanced accuracy) is well-predicted by our theory. We see that this is indeed the case for both class pairs in the supplement. We will ensure this point is clarified in the revision. We acknowledge that more extensive experimental results could strengthen the practicality of our work, and will gather results for different modalities (e.g. text) and different datasets to include in the revision.
> > >
> > > We hope that we have adequately addressed your concerns and enhanced your understanding of our work.

---

### Official Review · Reviewer_k7Vs · 2025-06-30

**Clarity:** 3
**Significance:** 3
**Originality:** 3
**Rating:** 4
**Confidence:** 1

**Summary:**

This paper investigate the loss weighting method in the understudied regime $\delta \in (0, 1)$, where $\delta$ is the ratio between the latent data dimension and the retraining samples. The authors theoretically and empirically show that the loss weighting is still effective in the regime $\delta \in (0, 1)$ and they also derive optimal asymptotic weighting as a function of $\delta$.

**Questions:**

- Would the results still hold under weaker assumptions than $d, n \rightarrow \infty$?

- How is the conclusion affected by the input complexity? The conclusions are derived under a special case where both p(x|Y = y+) and p(x|Y = y-) follow Gaussian distributions with identical variances. However, in real-world settings, different classes may exhibit varying input complexities, which can significantly impact the optimal choice of $\rho$. For instance, if the input x  corresponding to y+ is fixed to a single point, then selecting a small $\rho$ would be reasonable.


- Is there an overlap between $\delta \to 0$ and $\delta \in (0, 1)$? If so, what are the connections and differences between the conclusions drawn in this work and those in prior studies that focus on the $\delta \to 0$ regime?

**Ethical Concerns:**

["NO or VERY MINOR ethics concerns only"]

**Final Justification:**

My concerns are addressed and I will keep my original score

**Limitations:**

yes

**Paper Formatting Concerns:**

nil

**Quality:**

3

**Strengths And Weaknesses:**

I have limited knowledge in this specific area and the overall manuscript appears satisfactory to me.

Quality:

- [+] The theoretical and empirical demonstrations are both clear and technically sound.

Clarity:

- [+] The presentation of the idea is clear.
- [-] It is unclear why the case of $\delta = 1$ is special, especially considering that many dimensions in the last layer of large models often go unutilized. Further justification or intuition would be helpful.

Significance& Originality:

- [+] I think this work is novel and can be a contribution to the Machine learning community.
- [-] The current formulation is limited to the specific case where p(x|Y=y_i)is a Gaussian distribution with an identity covariance matrix. Generalization beyond this setting would enhance its impact.

---

> ### Author Rebuttal · Authors · 2025-07-30
>
> We thank the reviewer for their careful review of our work and are grateful for the opportunity to clarify these points. We will address each of the presented weakness and questions:
>  - **W1**: While many of the dimensions in the last layer go unused or have little impact, the threshold of $\delta=1$ is still critical in theory. At that point, the single optimizer of square loss is not well defined and theory must resort to discussing the minimum norm interpolator of the training data. This threshold, and in particular the regime of $\delta>1$,  is also used to argue against the efficacy of loss weighting in large models [1]. In contrast, we study the underparameterized regime which we demonstrate is amenable to loss weighting. We will clarify this point in an update to our manuscript if accepted.
>  - **W2**: The CGMT framework allows us to discuss more complex data distributions such as mixtures of Gaussians per class or more general covariance matrices. This work focuses on the identity covariance matrix for simplicity of presentation, but any shared covariance matrix is captured by our theory through a whitening transformation. We hope to extend this framework to weighting for mixtures of Gaussians per class, but this work is a first step towards that goal. Indeed we see that our analysis provides valuable insights in real datasets which are certainly non-Gaussian and non-isotropic. We will ensure that these clarifications are made in the revision.
>  - **Q1**: In our analysis of the auxiliary optimization, the only use of the asymptotics is through the Law of Large Numbers and kicks in relatively quickly. Indeed, we report results for n=20 and show that the theoretical weighting scheme is still effective despite the very small number of samples and dimensions.
>  - **Q2**: Indeed, one should suspect that the (relative) input complexity should also play a role in the selection of $\rho$. The central aim of our manuscript is to capture the effect of parameterization in the selection of loss weights, and we focus on the identity covariance as a clean basis for this investigation. We hope that this work serves as inspiration for future work exploring other factors that should be considered when selecting loss weight, including relative input complexity.
>  - **Q3**: One nice result of our work is that it captures the population case ($\delta\rightarrow0$) as a special case. We point out in [Section 3.1] that our weighting scheme $\tilde\rho$ recovers the ratio of priors which is optimal in the population setting. Additionally our results on downsampling in [Section 3.2] align with [2] when $\delta\rightarrow0$. Our results offer the flexibility to account for the parameterization ratio and are complementary to population results.
>
> We hope that these clarifications enhance the reviewer's understanding of our work and will ensure that these clarifications are included in the revision.
>
> ## References:
>
> [1] J. Byrd and Z. C. Lipton, “What is the Effect of Importance Weighting in Deep Learning?,” presented at the International Conference on Machine Learning, Dec. 2018.
>
> [2] M. Welfert, N. Stromberg, and L. Sankar, “Theoretical Guarantees of Data Augmented Last Layer Retraining Methods,” May 09, 2024, arXiv: arXiv:2405.05934. doi: 10.48550/arXiv.2405.05934.

---

> > ### Comment · Reviewer_k7Vs · 2025-08-05
> >
> > Thank the reviewer for addressing my concerns. I will keep my original score

---

### Official Review · Reviewer_odBN · 2025-07-01

**Clarity:** 3
**Significance:** 3
**Originality:** 3
**Rating:** 4
**Confidence:** 2

**Summary:**

- Study the effectiveness of loss weighting in LLR to improve  minority class accuracy, which is highly correlated to the overparameterization ratio $\delta$
-  Proves in theory and practice that loss weighting is still effective under the δ ∈ (0, 1) regime of LLR, but the optimal weight needs to consider the  overparameterization ratio $\delta$
- Provide a closed‐form solution for the square‐loss weights.
- Apply the method in the practical application of fine-tuning models.

**Questions:**

See weakness

**Ethical Concerns:**

["NO or VERY MINOR ethics concerns only"]

**Final Justification:**

The authors addressed my concern

**Limitations:**

Yes

**Paper Formatting Concerns:**

No concern

**Quality:**

3

**Strengths And Weaknesses:**

### Strengths: see the summary
### Weakness:
- The paper addresses only last‐layer retraining. While that setting is common, in many applications, a small number of top layers or additional modules are fine‐tuned jointly to improve performance in both sufficient and limited training data. Extending the analysis beyond one linear layer remains an open challenge.
- The closed‐form weighting is derived for the square loss. Although the scalar‐equation framework can apply more generally, the most commonly used classification loss, cross‐entropy, should be discussed.
- Strong assumption on the distribution of latent features being Gaussian with known covariances.

---

> ### Author Rebuttal · Authors · 2025-07-30
>
> We thank the reviewer for their careful consideration of our work. We will consider each of the weakness in turn:
>  - **W1**: The construction of the optimal weighting scheme $\tilde\rho$ relies on the linearity of the last layer retraining, but one could conjecture that the same trends would likely hold: namely the more parameters that are being finetuned (with respect to the number of samples and their dimension), the stronger of a weight is needed to achieve the same balance. A key challenge in settings rather than the last layer retraining would be quantifying the effective dimension as our current approach relies on a fixed latent distribution. Spreading weights across different layers would complicate this significantly.
>  - **W2**: A key challenge in characterizing losses other than the square loss is in computing the explicit form of the expected Moreau envelope; in other words, such closed form expressions do not exist for most losses making the optimization harder in theory. While the reviewer correctly points out that the scalarization holds for any loss, the reduction beyond that point is dependent on the form of the Moreau envelope. Recent work [1] has found that finetuning models using square loss can achieve equal or better classification performance to using standard cross entropy, which we discuss in the manuscript as motivation for studying the optimal weighting scheme for square loss.
>  - **W3**: This assumption may appear strong, but recent work [2,3] has argued that such assumptions in the last layer can be sufficient to guarantee performance for distributions sharing only the first and second order statistics. Additionally, while the nature of the CGMT framework requires that the latent features have some variant of a Gaussian distribution, there is the possibility of extending this work to mixtures of Gaussians which may more closely resemble the true latent distribution of a deep model. Finally, our empirical results align with our theoretical findings, suggesting that the assumptions we make might not be too strong to be useful in practice.
>
> ## References:
>
> [1] A. Achille, A. Golatkar, A. Ravichandran, M. Polito, and S. Soatto, “LQF: Linear Quadratic Fine-Tuning,” Dec. 21, 2020, arXiv: arXiv:2012.11140. doi: 10.48550/arXiv.2012.11140.
>
> [1] R. Ghane, D. Akhtiamov, and B. Hassibi, “Universality in Transfer Learning for Linear Models,” Feb. 23, 2025, arXiv: arXiv:2410.02164. doi: 10.48550/arXiv.2410.02164.
>
> [2] Q. Han and Y. Shen, “Universality of regularized regression estimators in high dimensions,” Jun. 27, 2022, arXiv: arXiv:2206.07936. doi: 10.48550/arXiv.2206.07936.

---

### Official Review · Reviewer_T2Wg · 2025-07-01

**Clarity:** 2
**Significance:** 2
**Originality:** 3
**Rating:** 4
**Confidence:** 3

**Summary:**

This paper studies optimal loss weighting in last layer retraining for the underparameterized regime ($\omega = d/n \in (0,1)$), a practically relevant setting between the well-understood population and overparameterized extremes. The authors employ the Convex Gaussian Minimax Theorem (CGMT) to reduce the high-dimensional weighted ERM problem to a tractable system of four scalar equations, deriving closed-form solutions for square loss and an optimal weighting formula that incorporates overparameterization effects beyond the classical ratio-of-priors. They validate their theoretical predictions on CelebA and CIFAR10 datasets.

**Questions:**

1. The analysis focuses on binary classification, but many real-world applications involve multi-class imbalanced problems. How would the theoretical framework extend to the multi-class case, and would the optimal weighting strategy change significantly?
2. The work specifically targets the last layer retraining, but what if it were applied to broader fine-tuning scenarios? Would the optimal weighting formula still hold?

**Ethical Concerns:**

["NO or VERY MINOR ethics concerns only"]

**Final Justification:**

Although several concerns have been resolved, the multi-class scenario still raises questions for me. The author's two explanations have not fully convinced me.

**Limitations:**

Please see the weakness and questions

**Quality:**

3

**Strengths And Weaknesses:**

Strengths:
1. The paper tackles the understudied underparameterized regime ($\omega \in (0,1)$) in cost-sensitive learning, which is precisely where real-world last layer retraining typically operates, bridging the theoretical gap between well-characterised extreme cases.
2. The authors construct a well-structured theoretical framework that progresses systematically from general principles to concrete applications, utilising sophisticated mathematical tools (CGMT, Moreau envelopes) to derive theoretical insights.

Weaknesses:
1. A key limitation is that the authors assume solution uniqueness in Theorem 1 without proof, creating a conditional statement where the essential premise remains unverified. Additionally, CGMT provides probabilistic bounds, yet Theorem 1 claims deterministic convergence without bridging this fundamental gap through concentration inequalities.
2. A gap exists between theoretical analysis and practical implementations; specifically, the theory assumes $n, d \to \infty$ but practical applications use finite samples (e.g., $n=20$) with no error bounds or convergence rates provided.

---

> ### Author Rebuttal · Authors · 2025-07-30
>
> We thank the reviewer for their thorough reading of our work. We are very glad that you found the paper relevant and engaging. We hope to address each of the weaknesses and questions mentioned.
>
>  - **W1**: Thank you for pointing this out. The key to showing uniqueness (and therefore convergence to the asymptotic discussed in this work) is to show the joint strict convexity-concavity of the objective in (37) in the supplement. For square-loss, we can take a direct approach by evaluating the *expected* Moreau Envelope (eME) functions in (37) leveraging that the square-loss is self-dual, hence its Moreau envelope is an easy-to-work-with quadratic. After easy calculation of the expectations over the (scalar) Gaussian random variables, and after combining the two individual eME functions (corresponding to majorities/minorities respectively) we can show that the combination (which is of the form ($\alpha^2+\gamma^2+b^2+1)/\lambda$) is jointly strictly convex (in $\alpha,\gamma,b,\lambda$) by direct evaluation of the Hessian. We note that strict convexity-concavity holds only when both eME terms are considered together, as each individual eME is not jointly strictly convex-concave. Beyond square-loss, we can follow the steps of App. A.6.1 in [1]. The only key change, following the “lesson” learnt by the explicit evaluation of the square-loss above, is to combine the two eME terms. We will include this analysis in our final version and update corollary 1 to reflect this. We will also clarify in Thm. 1 that the convergence to the deterministic parameters holds in-probability (as previous works e.g. [1,24 in manuscript]).
>  - **W2**: First, we highlight the fact that asymptotics kick in at relatively small dimensions and samples is a *positive* feature of our analysis. Non-asymptotic analyses are more challenging. While the CGMT is non-asymptotic in nature, the technical challenge comes in the analysis of the Auxiliary optimization. Recent literature [2,3,4] has made significant progress in this direction, but given the necessary technical overhead such an extension goes beyond our scope. We recognize the value of such non-asymptotic extension, which for example can yield confidence bounds and will certainly discuss this as current limitation and a possible direction for future work. For our main goal of re-weight optimization it is encouraging that direct optimization of the asymptotic formulas gives insights that match with our empirics.
>  - **Q1**: Recent work [5,6] has considered the multi-class case, and our scalarization (Theorem 1) would be amenable to these updated analyses. However, the analysis of the auxiliary optimization and the weighting scheme would need to change. For instance, it is not obvious that the multi-class extension of square loss would be amenable to the same closed-form analysis. Additionally, the optimal weighting scheme in our work is chosen such that $b=0$ which is enough to ensure that both classes are fairly classified. In the multiclass setting, however, this condition would be insufficient without significant additional structure on the class means and variances. We hope to extend our work to this setting in the future, and see this manuscript as an important and valuable step in that direction.
>  - **Q2**: The construction of the optimal weighting scheme $\tilde\rho$ relies on the linearity of the last layer retraining, but one could conjecture that the same trends would likely hold: namely the more parameters that are being finetuned (with respect to the number of samples and their dimension), the stronger of a weight is needed to achieve the same balance. A key challenge in settings rather than the last layer retraining would be quantifying the effective dimension as our current approach relies on a fixed latent distribution. Spreading weights across different layers would complicate this significantly.
>
> ## References:
>
> [1] H. Taheri, R. Pedarsani, and C. Thrampoulidis, “Sharp Asymptotics and Optimal Performance for Inference in Binary Models,” Feb. 26, 2020, arXiv: arXiv:2002.07284. doi: 10.48550/arXiv.2002.07284.
>
> [2] L. Zhou, F. Koehler, P. Sur, D. J. Sutherland, and N. Srebro, “A Non-Asymptotic Moreau Envelope Theory for High-Dimensional Generalized Linear Models,” Oct. 21, 2022, arXiv: arXiv:2210.12082.
>
> [3] G. Wang, K. Donhauser, and F. Yang, “Tight bounds for minimum $\ell_1$-norm interpolation of noisy data,” in Proceedings of The 25th International Conference on Artificial Intelligence and Statistics, PMLR, May 2022, pp. 10572–10602.
>
> [4] Q. Han and Y. Shen, “Universality of regularized regression estimators in high dimensions,” Jun. 27, 2022, arXiv: arXiv:2206.07936.
>
> [5] B. Loureiro, G. Sicuro, C. Gerbelot, A. Pacco, F. Krzakala, and L. Zdeborová, “Learning Gaussian Mixtures with Generalised Linear Models: Precise Asymptotics in High-dimensions”.
>
> [6] D. Bosch and A. Panahi, “A Novel Convex Gaussian Min Max Theorem for Repeated Features”. Proceedings of The 28th International Conference on Artificial Intelligence and Statistics

---

> > ### Comment · Reviewer_T2Wg · 2025-08-05
> > **Response**
> >
> > I appreciate the authors' explanations. The discussions of the solution strategy and related work have partially addressed my concerns. However, as other reviewers have noted, the lack of consideration for multi-class classification problems and constraints in the loss formulation creates a concerning gap between the theoretical framework and practical applications. I will maintain the original score.

---

> > > ### Author Response · Authors · 2025-08-08
> > >
> > > We are glad to have addressed your concerns regarding the analysis of the binary objective and we thank you for your continued engagement.
> > >
> > > We agree that the multi-class setting is of great practical interest, but this introduces significant new challenges in the analysis as we have pointed out. As mentioned above, the key challenge is the optimization of the loss weights, of which there would be $m-1$ in the $m$-class classification case. Even in the binary case there is a dual purpose that this weight must serve: balancing the intercept ($b$) and aligning the classifier with the means ($\alpha$). The key observation that makes the binary analysis achievable is that improvements in alignment ($\alpha$) benefit both classes (for reasonable intercept) and the intercept $b$ trades off between classes. In the $m$-class setting, this intuition no longer holds directly: alignments of a classifier with one class can hurt their performance on other classes. This, coupled with the number of weights to optimize, is the driver of the complexity of the multi-class analysis.
> > >
> > > We strongly believe our binary analysis can give us some intuition for the behavior of weighted multi-class classification. From our analysis, we should expect that for large $n$ the standard ratio of the priors is nearly optimal, while for small $n$ the weights should begin to favor the smallest class. We will add empirical results exploring this hardening effect to the revision of our manuscript as a means of extending the intuition our analysis provides to the multi-class setting.

---

### Decision · Program_Chairs · 2025-09-17

**Decision:**

Accept (poster)

**Comment:**

"Thumb on the Scale" studies optimal loss weighting for last layer retraining, showing that it is still effective in the underparameterized regime if weights take relative overparameterization into account. Reviewers generally appreciated the work, acknowledging that it constructs a well-structured theoretical framework, and finding it to be a solid contribution to the machine learning community. While the study is limited to binary classification, and Reviewer T2Wg finds the extension to the multi-class scenario to be unconvincing, there is still sufficient material here to justify acceptance in my view. I urge the authors to include any elaborations provided during the discussion phase in the camera-ready submission.